# On the Vertically Stacked Gate-All-Around Nanosheet and Nanowire Transistor Scaling beyond the 5 nm Technology Node

**DOI:** 10.3390/nano12101739

**Published:** 2022-05-19

**Authors:** Hei Wong, Kuniyuki Kakushima

**Affiliations:** 1Department of Electrical Engineering, City University of Hong Kong, Hong Kong, China; 2Frontier Research Center, Tokyo Institute of Technology, Yokohama 152-8550, Japan; kakushima.k.aa@m.titech.ac.jp

**Keywords:** vertically stacked nanosheet transistor, FinFET, nanowire transistor, channel width folding, nano CMOS

## Abstract

This work performs a detailed comparison of the channel width folding effectiveness of the FinFET, vertically stacked nanosheet transistor (VNSFET), and vertically stacked nanowire transistor (VNWFET) under the constraints of the same vertical (fin) height and layout footprint size (fin width) defined by the same lithography and dry etching capabilities of a foundry. The results show that the nanosheet structure has advantages only when the intersheet spacing or vertical sheet pitch is less than the sheet width. Additionally, for the nanowire transistors, the wire spacing should be less than 57% of the wire diameter in order to have a folding ratio better than a FinFET with the same total height and footprint. Considering the technological constraints for the gate oxide and metal gate thicknesses, the minimum intersheet/interwire spacing should be in the range of 7 to 8 nm. Then, the VNSFET structure has the advantage of boosting the chip density over the FinFET ones only when the sheet width is wider than 8 nm. On the other hand, the VNWFET structure may have a better footprint sizing than the FinFET ones only when the nanowire diameter is larger than 14 nm. In addition, considering the different channel mobilities along the different surface directions of the silicon channel and also some other unfavorable natures such as more complicated processes, more significant surface roughness scattering, and parasitic capacitance effects, the nanosheet transistor does not show superior scaling capability than the FinFET counterpart when approaching the ultimate technology node.

## 1. Introduction

To suppress the downsizing-induced short-channel effects, such as the threshold voltage roll-off, subthreshold conduction, and drain-induced barrier lowering (DIBL), the MOS device structure has evolved from a single-gate structure to a double-gate (DG) structure, and then to the state-of-the-art tri-gate FinFET structure [1,2,3,4,5,6,7,8]. The short-channel effects were suppressed by increasing the channel doping and thinning the body, such as silicon-on-insulator (SOI), and more recently by introducing multi-gate structures for a better gate electrostatic control [1,2,3,4,5,6,7,8,9]. The effectiveness of the gate electrostatic control can be modeled by the electrostatic integrity (*EI*) or the natural length as derived by solving the channel potential from Poisson’s equation [8,9]. The natural length decreases as the number of gates increases [9]. From this point of view, a gate-all-around nanosheet or nanowire structure has even better short-channel suppression capability than the currently used FinFET structure and is considered to be a better option for the next technology advancement [8,9,10,11,12,13,14,15,16,17]; although, the actual gate length of both FinFET and GAA nanosheet/nanowire may both stall at around 6 to 8 nm due to the limitation of EUV lithography [18].

The advantage of the multi-gate structure is not limited to the aforementioned better electrical characteristics but also due to its smaller footprint or layout size for the same drain current level and thus leading to the increase of the chip density in the pace of Moore’s Law although the gate-length shortening rate has been slowed down when approaching the lithography limit [14,15,18]. This, in effect, this results in a better technology similar to the effect of gate-length shortening, and thus a shorter technology node number such as 7 nm and 5 nm technology nodes were assigned to the better chip, although the actual physical gate lengths are in fact much longer [18].

There was a consensus that the vertically stacked gate-all-around (GAA) nanosheet field-effect transistor (VNSFET) structure should be the candidate, yet the last MOS structure, for technology nodes beyond 5 nm [14,15]. Unlike the FinFET technology, in which the gate surrounds the channel on three sides to help control current leakage, the gate in the VNSFET technology completely surrounded the channel to provide the best electrostatic control of the short-channel effects. It was shown that the natural length of the GAA nanosheet is about 30% smaller than the double-gate structure [14]. An even more attractive feature is that, by stacking the GAA channel vertically, the drain current or the effective channel width could be much larger than a FinFET with the same footprint and thus has even better scalability. In the 12 nm long gate vertically- stacked nanosheet transistor demonstrated by a joint effort of IBM, Samsung Electronics, and Globalfoundries [15], the nanosheet transistor was found to have much better performance than a FinFET with the same footprint or fin width for most of the performance indicators explored. The chip fabricated with this new process consumes 75 percent less energy as compared with the “7 nm” chips. It is rather attractive that better performance and higher chip density can be achieved with relatively less advanced lithography technology. It was further suggested that the VNSFET can help to scale the CMOS technology down to 2 nm, and it is anticipated that this structure is likely to be the last step of Moore’s Law in device downsizing [14,15]. It was also suggested that the FinFET structure is no longer suitable for 7 nm technology and beyond which is shown not to be the case. In addition, it is noted that the device fabrication process was based on less advanced technology, with a gate length of 12 nm, fin width of 50 nm, and fin high of less than 50 nm [15]. It can be considered as realizing a 7 nm chip with the 10/14 nanometer facility/process. However, the performance was not compared with the most advanced 5 nm FinFET technology in manufacturing. The conjecture that the nanosheet technology has better scalability than the FinFET counterpart is yet to be explored. From the geometry point of view, the effective channel width of the nanosheet may not always be larger than the FinFET as far as the same footprint is concerned. Fin width, fin height, and intersheet spacing are the key limiting factors. On the other hand, a nanosheet should have a smaller cross-section and a larger surface region, which is found to cause significant mobility degradation [17,19,20,21]. Meanwhile, it was reported that the 3 nm fab constructed by TSMC will still be based on FinFET technology. If this is the case, there may be only one generation difference between these two technologies. It is worth having a detailed investigation of both technologies under the same constraints. It would have great technical value in exploring the ultimate scalability of both technologies. The objective of this work intent to conduct a comprehensive comparison of these two technologies based on certain technological constraints.

In this work, we shall perform a comprehensive study on the effective channel width of FinFET, VNSFET, and vertically stacked nanowire transistor (VNWFET) with the same confinement of photolithography-defined layout footprint, and the etching-governed vertical height or headroom of the silicon nanostructure. Nanoscale photolithography and deep dry etching are major constraints of nanofabrication technology. In Section 2, we shall first compare the geometry folding ratio of these three transistor structures. We shall show that the VNWFET structure should have no advantage over the FinFET and VNWFET. For the VNSFET structure, it does show up an attractive folding ratio, but it is limited to a large footprint and a short fin height only. In Section 3, we further consider the different mobilities associated with the channel crystalline orientations. It shows that the advantage of the VNSFET over the FinFET is still limited to a footprint of over 10 nm. Beyond this size, VNSFET should have more significant mobility degradation because of the more pronounced contribution of surface roughness scattering. This issue will be manifested by the experimental findings to be presented in Section 4. We shall further elaborate on some other possible drawbacks associated with the VNSFET structure in Section 5. Finally, we highlight the major findings of this work in Section 6.

## 2. Width Folding and Moore’s Law Scaling

It is noted that the technology node number has been decoupled from the gate length or half-pitch width of the device size since the introduction of the three-dimensional FinFET structure in the 22 nm technology node. For example, the minimum gate length in the latest 5 nm technology node is estimated in the range of 8 to 12 nm [18]. In the tri-gate FinFET structure, and so as in the GAA nanosheet or nanowire structure, a larger effective channel width is possible with the same footprint as in the double-gate or single-gate structure. In addition to the gate-length scale, the effective gate width, under the constraint of fin width, is folded around the three sides of a fin and four sides of a nanosheet and multiplied by the number of nanosheets stacked vertically in VNSFET. As a result, the chip density, the key technological indicator of Moore’s law, can be doubled even if the gate length reduction is no longer followed by the 0.7× rule. It can be considered as more advanced technology, or the chip is implemented with a smaller size transistor or with a smaller number of the technological node when compared to the conventional 2D device structure. In addition, the recent technology node numbers, announced by different silicon foundries, are now not directly comparable [18]. For the same node number, the performance indicators of different foundries could be quite different. However, a smaller node number is still meaningful for comparing the performance or the chip density for different generations of processes of the same foundry. A smaller node number indicates a higher chip density and that is a result due to both the gate-length shortening and gate-width footprint reduction by folding the gate width around three sides of FinFET and increasing the fin height. In this section, we conduct a simple derivation on the effective channel width of FinFET, VNSFET, and VNWFET and then compare the effectiveness in folding ratio improvement under the same footprint (or mask layout size achieved by the same lithography process, and headroom (fin height producing by the similar dry-etch process).

Figure 1 compares the cross-sections of FinFET, vertically stacked GAA nanosheets, and vertically stacked GAA nanowire structure. Figure 2 shows the current–voltage characteristics of a typical FinFET and a single-level nanosheet transistor with the same footprint of 25 nm and headroom of 40 nm. Both transistors have a top (horizontal channel in <100> direction and the sidewall channels in the <110> direction. Because of the larger effective channel width and the advantage large portion of the horizontal channel with higher effective mobility, the nanosheet transistor shows a significantly larger drain current. The drain current is 1.6 times at V_GS_ = 2.5 V and V_DS_ = 4 V. Three issues can be derived by a quick evaluation of the characteristics and structures of both devices:The footprints and thus the horizontal channel of most reported nanosheet FETs are usually much larger than the most advanced FinFET technology. Does the advantage remain if the footprint is reduced to 10 nm and smaller?The comparison usually takes advantage of nanosheet transistors where the high mobility portion is assigned to larger horizontal parts of the nanosheet channel. In contrast, high mobility is assigned to the negligible top channel region in the FinFET structure. Is the advantage still maintained if the high mobility is re-allocated to the sidewall of a tall fin FinFET?Mobilities are found to be significantly degraded due to the significant contribution of the surface region to the channel conduction. What is this effect on the FinFET and VNSFET scaling?

We are going to answer these questions with experimental findings and some careful theoretical modeling.

### 2.1. FinFET Width Folding Ratio

As shown in Figure 1a, the current conduction in the FinFET is not only via the top surface; it also flows over the two vertical sidewalls of the fin. For a FinFET with fin height and fin width of *H_fin_* and *W_fin_*, respectively, the effective channel width of the FinFET is
(1)$$W_{eff,fin}=W_{fin}+2H_{fin}$$

Assuming the width of the mask layout or the footprint equals to fin width, then the effective sizing or the folding ratio of the channel width can be calculated as
(2)$$Folding\;Ratio=\frac{W_{fin}}{W_{eff}}=\frac{W_{fin}}{W_{fin}+2H_{fin}}\;$$

This parameter represents the capability of squeezing a large electrical channel width, $W_{eff,fin}$, into a small layout width, *W_fin_*. A smaller folding ratio enables a higher chip density, and it represents a more advanced technology. That is, the physical size of the transistor from the two-dimensional point of view is a factor of $W_{fin}/(W_{fin}+2H_{fin})$ smaller. The higher the fin, the smaller the folding ratio. As mentioned, since the 22 nm technological node, the higher chip density for each generation was achieved both by shortening the gate length and reducing the folding ratio. As given in (2), the smaller folding ratio of FinFET can be achieved by increasing the fin height. That is the reason why the 7 nm and 5 nm technology claimed by the foundries actually have a gate length longer than 10 nm.

### 2.2. Width Folding Ratio of Vertically Stacked Nanosheet FET

GAA structure, in particular, the VNSFET had been considered to be the ultimate scalable structure for the MOS device evolution. It has even better electrostatic integrity than the FinFET ones. IBM recently suggested the vertically stacked nanosheet gate-all-around structure (VNSFET) should be the last resolution for the ultimate MOS device scaling [14,15]. For the nanosheet structure shown in Figure 1b, if the sheet height is *h_s_*, then the perimeter of a nanosheet cross-section or the effective channel width of the sheet will be
(3)$$W_{s}=2(W_{fin}+h_{s})$$

For *n* nanosheets stacked together with a bottom bulk channel, the total effective width is
(4)$$W_{eff,ns}=W_{fin}+2n(W_{fin}+h_{s})$$

Considering the spacing between the sheets is *αh_s_*, the achievable total etch depth or fin height produced by a certain etching process is *H_fin_*, then the total number of nanosheets is
(5)$$n=\frac{H_{fin}}{h_{s}\left(1+\alpha \right)}$$

Here, *n* should be truncated into an integer and the value may be adjusted according to the fin height and sheet height. If we assume all these parameters together can make an integer number *n*, combing Equations (4) and (5), yields
(6)$$W_{eff,ns}=W_{fin}+\frac{2H_{fin}}{1+\alpha }\left(1+\frac{W_{fin}}{h_{s}}\right)$$

It is noted that the effective channel width given in Equation (6) depends on the sheet height and intersheet spacing. Larger spacing results in a larger portion of the vertical sidewall not being used as the conduction channel, which may discount the merit of the four-side channel of the nanosheet. In order to have a larger effective width than a FinFET with the same fin width and total height, i.e., $W_{eff,ns}>W_{eff,fin}$*,* we have
(7)$$W_{fin}+\frac{2H_{fin}}{1+\alpha }\left(1+\frac{W_{fin}}{h_{s}}\right)>W_{fin}+2H_{fin}$$

Solving Equation (7) for *α*, we have
(8)$$\alpha <\frac{W_{fin}}{h_{s}}$$

Equation (8) indicates that the nanosheet structure should have a better folding factor if the spacing between the nanosheets is smaller than the fin width. The minimum spacing is limited by both material property and the fabrication process. It has been reported that half nanometer EOT is technically feasible with a 3- to 4-nm-thick high-k oxide [2,22,23]. High-k oxide thinner than 3 nm would result in a significant direct tunneling current [2,22] and other parameter variability issues resulting from the surface roughness [23], which are not acceptable in the giga-scale circuit from the power dissipation and parameter variability points of view. Let us consider the physical thickness of gate oxide to be 3 nm, and the metal gate can be reduced to 2 nm. Then the intersheet/interwire spacing, *αh_s_*, should not be less than 8 nm. It must be pointed out that here 8 nm is already the most optimistic estimation; it may not be a viable option in mass production. Nevertheless, under this assumption, a VNSFET is better than a FinFET structure only when the sheet width, *W_fin_*, is larger than 8 nm. That is, the nanosheet technology does not have much advantage when compared with the state-of-the-art EUV-based 5 nm or 7 nm technologies. It has advantages only for some foundries which are not able to pattern a smaller fin width and to produce a tall fin.

### 2.3. Width Folding Ratio of Vertically Stacked Nanowire FET

For the vertically stacked nanowire GAA FET shown in Figure 1c, we can perform a similar calculation. Considering the nanowire is round in the cross-section, the effective channel width of a single nanowire is
(9)$$W_{wire}=\pi W_{fin}$$

It is noted that it is not uncommon that the cross-section of the resultant nanowire is not round [13]. If it is the case, then the width will be smaller than $\pi W_{fin}$. For the sake of simplicity, here we assume the nanowire is a perfect cylindrical in shape. For *n* nanowires with a bottom bulk flat channel, as shown in Figure 1c, the total effective channel width is
(10)$$W_{eff,nw}=W_{fin}+n\times \pi W_{fin}$$

Considering the spacing between the nanowires is *αW_fin_*, the fin height produced by the process is *H_fin_*, then the total number of nanowires is
(11)$$n=\frac{H_{fin}}{W_{fin}\left(1+\alpha \right)}$$

Combining Equations (10) and (11), yields
(12)$$W_{eff,nw}=W_{fin}+\frac{\pi H_{fin}}{1+\alpha }$$

Again, to have a better folding ratio than a FinFET with the same height and same width, i.e., $W_{eff,nw}>W_{eff,fin}$, or
(13)$$W_{fin}+\frac{\pi H_{fin}}{1+\alpha }>W_{fin}+2H_{fin}$$

Solve Equation (13) for *α*, we have
(14)$$\alpha <\frac{\left(\pi -2\right)}{2}\approx 0.57$$

That is, to take advantage of the nanowire structure, the spacing between the nanowire should be less than about 57% of the nanowire size (diameter). Again, considering the limit of gate oxide thickness and the metal gate thickness together of 8 nm, the VNWFET structure has a benefit only for the process with a fin width larger than 14 nm.

To have a more intuitive view of the effects of various parameters. We compare in Figure 3 the folding ratio of various FinFETs, vertically stacked nanosheets, and vertically stacked nanowire transistors. In Figure 3a,b, we assume the fin height is 80 nm. This fin height should be feasible and will be likely to be adopted for manufacturing in some future technological nodes [24]. For the FinFET structure, the folding ratio ranges from about 0.03 to about 0.087 for fin width increasing from 5 nm to 15 nm. In Figure 3a, we assume the sheet height is 6 nm. In the nanosheet transistor reported by the IBM group, the sheet height is reduced to 5 nm. Yet 5 nm thick could be the ultimate scale for nanosheet technology from the geometrical point of view. It was reported that there is a significant reduction in electron mobility when the silicon nanowire is reduced to 5 nm [25]; then, the benefit of scaling will be significantly discounted. Thus, we prefer to use 6 nm as the lower bound of nanosheet height. If the 80 nm headroom can accommodate 5 nanosheets, then the intersheet spacing is 10 nm, the same value reported by the IBM group [15]. Under this situation, the *α* value is 1.67 and the nanosheet structure is better than FinFET only when the fin width is larger than 10 nm. If we are going to squeeze 6 nanosheets or even 7 nanosheets with the same sheet height of 6 nm into the 80 nm tall fin, then the intersheet spacing needs to be reduced to about 7 nm and 5.4 nm, respectively. That requires the gate oxide thickness to be reduced to less than 2.5 nm and that technology has not been reported. Even if the technology is possible, VNSFET technology has the advantage of a better folding ratio only when the fin width is larger than 7 nm and 6 nm, respectively. Yet a more practical option based on the present technology is to use an 8-nm-thick nanosheet. Figure 3b shows the case of 8-nm thick for 4, 5, and 6 nanosheets incorporated into the 80-nm-tall fin. The *α* values are 1.2, 1.0, and 0.67 which correspond to the intersheet spacing of 12 nm, 8 nm, and 5.3 nm, respectively. As mentioned, the 5.3 nm seems not to be practical from the technical point of view. For 12 nm and 8 nm spacing, they should be achievable and the VNSFET structure is preferable only when the fin width is larger than 12 nm and 8 nm, respectively. Hence, the nanosheet technology could be a cost-effective option for foundries that are not able to pattern a fin structure smaller than 8 nm. Yet, 80-nm fin height is not very mature to certain foundries, for 60-nm-tall fin height (see Figure 3c), the sheet height between 6 to 8 nm and intersheet spacing of 7 to 9 nanometers could be an achievable technology. Then, the nanosheet structure does show an advantage for fin width larger than 7 or 9 nm.

Figure 3d compares the folding ratio of vertically stacked nanowire transistor with FinFETs with the same tall of 80 nm at different widths. Here, we assume the nanowire size is the same as the fin width. The number of nanowires being able to squeeze into the headroom given by the fin height is governed by wire size and the interwire spacing as well. For a wire size of 5 nm, we can produce 5 to 7 wires depending on the spacing. However, they are no advantage when we compare it with the FinFET. In fact, in only the case of wire size of 14 nm, 4 wires with an interwire spacing of 6 nm, can result in a notable improvement over the FinFET ones. This spacing is so small that it is not feasible with the present gate dielectric technology. Hence, from the size scaling point of view, a vertically stacked nanowire structure is not a favorable option over the FinFET ones; although, it has the best short-channel effect immunity among all the device structures being invented.

## 3. The Effect of Channel Mobility on Scaling

In addition to the width folding, channel mobility is another parameter that affects the strategy in choosing the device structure and size scaling. For achieving the same current value, the channel width can be smaller if the channel mobility is larger. In addition, from the operation speed point of view, larger mobility produces the same effect as shorter channel length. However, mobility optimization can also be achieved according to the channel structure if the channel length cannot be shortened. In the 3D FinFET and VNSFET, considering the mobility values are different in different crystal orientations and are size dependent, there are technical values to compare the effective mobility of FinFET and VNSFET [17,19,20,21,25].

A simple and meaningful comparison, based on the elementary drain current relationship, is to calculate the effective mobility-effective channel width product (MWP) of the device, i.e., $\mu_{eff,fin}W_{eff,fin}$. The excellent folding ratio of FinFET arises from the small fin width and tall fin height. That means the most significant conduction paths are contributed by the sidewall. Thus, to have a better performance, the FinFET should be so designed that the sidewalls should have the highest mobility, *µ_H_*, and the top surface can have a smaller mobility, *µ_L_*, (see Figure 4a). The mobility-width product can be expressed as
(15)$$\mu_{eff,fin}W_{eff,fin}=\mu_{L}W_{fin}+2\mu_{H}H_{fin}$$

A good FinFET should have $W_{fin}\ll H_{fin}$, and for the sake of simplicity, we assume $\mu_{H}\approx 2\mu_{L}$. Equation (15) can thus be approximated by
(16)$$\mu_{eff,fin}W_{eff,fin}\approx 4\mu_{L}H_{fin}+2\mu_{L}W_{fin}$$

For a VNSFET, the effective width is mainly contributed by the sheet width along the horizontal direction. The sheet should be so designed that the highest mobility is put on the width direction and the low mobility can be assigned to the sheet height direction (see Figure 4b). For a vertically stacked nanosheet transistor with *n* sheets, the mobility-width product is
(17)$$\mu_{eff,ns}W_{eff,ns}=n\left(2\mu_{H}W_{fin}+2\mu_{L}h_{s}\right)+\mu_{H}W_{fin}$$

As shown in Equation (8), a good nanosheet should have $W_{fin}\gg h_{s}$, and assuming $\mu_{H}\approx 2\mu_{L}$ also, Equation (17) can be approximated by
(18)$$\mu_{eff,ns}W_{eff,ns}\approx (4n+2)\mu_{L}W_{fin}+\frac{2}{1+\alpha }\mu_{L}H_{fin}$$

Thus, to achieve a better MWP than FinFET, i.e.,
(19)$$(4n+2)\mu_{L}W_{fin}+\frac{2}{1+\alpha }\mu_{L}H_{fin}\;>\;4\mu_{L}H_{fin}+2\mu_{L}W_{fin}$$

Solving Equation (19) leads to
(20)$$n>\frac{\alpha +0.5}{\alpha +1}\frac{H_{fin}}{W_{fin}}$$

Equation (20) indicates that the comparative advantage of VNSFET is governed by the fin width-to-fin height aspect ratio and the intersheet spacing parameter *α*. As mentioned, the spacing is predominately governed by the minimum achievable gate dielectric thickness. If we consider the minimum sheet height as 6 nm and the minimum spacing as 8 nm, then *α* ≈ 1.33. If the spacing is equal to the sheet height, then *α* = 1. A rough but yet reasonable approximation of Equation (20) is
(21)$$n>\left(0.75\sim 0.79\right)\frac{H_{fin}}{W_{fin}}$$

Equation (21) indicates that when the *H_fin_*/*W_fin_* is small, namely smaller fin height and wider fin width, a VNSFET can easily surpass the FinFET by implementing multiple vertically stacked channels. For larger a *H_fin_*/*W_fin_* value, it requires a larger number of nanosheets which is not feasible considering the headroom and the minimum intersheet spacing as defined by the possible thinnest gate oxide.

Figure 5a compares the 60-nm-tall FinFET with a VNSFET of the same height. For FinFET, since the effective width is mainly contributed by the sidewalls, changing the fin width has little effect on the effective mobility-width product value. If the number of sheets is 3, the VNSFET shows to have an advantage (larger effective mobility-width product) for sheet width larger than 14 nm. This breakeven point agrees with what was predicted in Equation (21). If the sheet height is reduced to 8 nm, and 4 nanosheets can be incorporated into the 60-nm headroom, then the VNSFET has a better MWP for sheet width larger than 10 nm. When *W_fin_* = 10 nm, *H_fin_*/*W_fin,_* = 6, and *n* ≈ 4 according to Equation (21). That agrees with this plot. If a more advanced 80-nm-tall fin can be etched, the nanosheet height can be further reduced to 6 nm, and the intersheet spacing can be extremely scaled down to 7 nm, then the breakeven point of nanosheet structure is still stalled at 10 nm (See Figure 5b). If the process can incorporate 5-nm-thick sheets into the 80 nm headroom, then the breakeven point is about 12 nm. These results again suggest that the vertically stacked nanosheet structure is favorable only to some less-advanced fabrication processes. It could be an attractive option for pushing the existing 10 nm or 14 nm technology a couple of generations further and that is what has been demonstrated in the literature [15]. However, it does not have much benefit for the foundries that already achieve the 5 nm FinFET-based technology with fin height greater 60 nm and footprint/fin width of about 10 nm. Nevertheless, the VNSFET structure is not likely a technological option for 3 nm technology and beyond.

## 4. Effect of Mobility Nanosheet and Fin Sizing

It was reported that the nanosheet and nanowire transistor usually have smaller channel mobility than the conventional ones [19,20,21,25]. Additionally, the mobilities are also found to be size-dependent. Here, we further conducted comprehensive experimental measurements on the sizing effect of the nanowire transistor. The nanowire transistors were fabricated in Iwai’s Lab at the Tokyo Institute of Technology, Tokyo, Japan. The starting material was an SOI wafer with a buried oxide (BOX) layer. Silicon fins of about 15 nm high with widths varied from 10 nm to 130 nm were patterned using a silicon nitride mask. The source and drain (S/D) regions were patterned with a thermally grown oxide layer and their dopings were done with phosphorus implantation. After the post-implant annealing at 800 °C, the TiN gate electrode was deposited and patterned. To avoid possible short-channel effects, the devices used in this investigation had a long gate length of 2 μm.

Figure 6a,b compare the temperature dependence of nanosheet FET with different sheet widths. Figure 6a,b further show that the temperature dependence is governed by the sheet width also. As shown in Figure 6a, for a sheet with 10 nm width, the I-V characteristics only have a small change. For 25 nm sheet-width device characteristics shown in Figure 6b, the drain current is significantly enhanced for temperature increasing from 300 K to 400 K. Figure 6c plots and drain current as a function of sheet width at room temperature. Figure 6d plots the drain current as a function of temperature for different fin widths. Large temperature dependence is found for larger fin width. This result can be attributed to the more significant contribution of the surface roughness scattering in the surface region. The mobility of semiconductors is governed by coulomb, phonon, and surface roughness scattering. The coulomb scattering event is inversely proportional to the temperature whereas the phonon scattering is proportional to temperature and the surface roughness scattering is temperature independent. According to these facts, it is expected that the drain current should increase with the temperature because of the more significant contribution of phonon scattering at T > 300 K. However, the temperature increasing rate is lower as the fin width becomes smaller and the current almost remains unchanged for *W_fin_* = 10 nm. This trend can be explained by the increasing contribution of surface roughness scattering in narrower sheets. For *W_fin_* = 10 nm, a significant drop in drain current is found (see Figure 6c). It suggests that the current conduction mainly via the surface region and mobility degraded significantly due to increased surface roughness scattering. It is further noted that at small V_DS_ < 1 V, especially for sheet width of 10 nm as shown in Figure 6a, sub-linear I_D_-V_DS_ relationships were found. This observation should be due to the more predominant effects of the surface roughness scattering and surface defect trapping which should result in a smaller channel mobility and a smaller amount of charge carrier in the channel. For the larger sheet width of 25 nm shown in Figure 6b, this phenomenon is alleviated. According to these observations, one may suggest that silicon nanosheets or nanowires with sizes less than 10 nm would experience a significant mobility degradation and should be special care when considering the size reduction. Yet the surface roughness can be improved with better etching and gate oxide deposition techniques. The amount of surface roughness scattering may still be acceptable for nanowire or nanosheet with thickness or width of 6 to 8 nm. From this point of view, one can again deduce that FinFET should have less surface roughness scattering induced mobility degradation as the size of the fin should be larger than the nanowire or nanosheet regarding the same footprint and same headroom are concerned. The surface region to bulk silicon ratio in a FinFET is smaller. It was reported that the mobility of FinFET is 2.5 times larger than the nanowire with the same width [21].

## 5. Discussions

We have demonstrated that the advantage of the VNSFET in terms of scaling will be dismissed if the footprint is smaller than 10 nm. For a footprint larger than 10 nm, the VNSFET does provide a large drain current under the same footprint as in the corresponding scale of FinFET. It should be an attractive technology option for upgrading the existing production line to a more advanced technology node without introducing a more advanced lithography machine. The calculations also show that further downsizing of VNSFET is governed by the thickness of the nanosheets. The nanosheet preparations are still based on the anisotropic etching of the Si_x_Ge_y_ superlattice. The etching rate can be controlled by varying the composition and with different etching techniques [19,20,21,26,27,28]. Worth mentioning achievement is the seven-level nanosheet structure reported on a 150-nm-thick Si_0.7_Ge_0.3_/Si multilayers [28]. However, it is still based on a footprint larger than 15 nm and the nanosheet thickness was kept at 8 nm; in addition, the intersheet spacing seems to be not able to reduce further. This improved VNSFET fabrication process demonstrates lots of improved performances over the previous three-level NSFET. It is noted that the improvement is not only due to the increased level of nanosheet stack but to the amazing etching technique being developed. Without disclosing the details of the etching technique and equipment used, Barraud et al., demonstrate a straight and tall fin of 150 nm [28]. The widths of the first to the seventh nanosheets are almost identical, and the surface and sidewalls of the sheet look uniform and smooth. It is not clear if the same process can be applied to the FinFET etching or not. If the 150-nm-tall fin is possible with such a nice sidewall morphology, the FinFET structure should still have a better folding ratio for a smaller footprint. The main reason that a VNSFET could not have a better folding ratio than the FinFET with the same footprint is the limit of the intersheet spacing of VNSFET. Eight nanometer-thick can be a feasible figure and the ultimate limit should not be smaller than six nm. Another factor that would have unfavorable consequences for the VNSFET scaling is the larger parasitic capacitance arising from the fringing fields along the nanosheet edges and the larger intersheet capacitances [29]. These parasitic capacitances together result in a more significant charging current and reduce the effect of gate capacitance. It further reduces the operation speed of the device. This issue is worth a comprehensive study.

Based on the study presented here together with the technological constraint of gate dielectric engineering and dry etching, we are inclined to believe that vertically stacked nanosheet structure is not a favorable option for “3 nm technology” and beyond. FinFET should still be a sustainable option that can be maintained by further increasing the aspect ratio (fin height-to-fin width ratio). Figure 7 shows the trend of the technology nodes since the introduction of the FinFET structure or 22 nm node [18]. Moore’s law in the chip density upscaling rate, taking Apple mobile CPU as an example, is still maintained so far. The increase in chip density does rely not only on the gate-length shortening, it is also achieved with the increased gate-width folding. The gate-length reduction rate had been slowed down to about 0.88 to 0.9 instead of the historical rate of about 0.7 per generation. For Intel’s “22 nm” technology, the actual gate length is around 25 nm. Additional scaling toward the “22 nm” specification was contributed by the gate-width folding. A better folding ratio was obtained by folding ratio by increasing the fin height and reducing the fin width. When approaching the EUV lithography limit, it is expected that the pace of gate length-shortening will go even slower, but we expected that Moore’s law will still maintain for a couple of generations. The gate-length scaling may be stalled at 6 nm or 8 nm not only because of the limit of EUV but also because of the effect of surface roughness scattering, as mentioned. If it is the case, the gate-length scaling shall be slower than the present rate (see the square markers of the gate-length line) and the product of the gate-length-folding ratio will be larger (see the triangular markers of the gate-length-folding ratio product curve). The smaller footprint can be achieved with a taller fin. A fin at 80 nm tall will be introduced in a couple of years. The ultra-tall fin of 130 nm may also be possible and “2 nm”, and “1 nm” technology can be achieved by scaling both the fin width and gate length to 6 to 8 nm.

## 6. Conclusions

We have conducted an intuitive and comparative analysis on the scaling issue of FinFETs and vertically stacked nanosheet transistors, and vertically stacked nanowire transistors beyond the 5 nm technology node. By calculating the folding ratio of the effective channel width of the three structures under the constraints of the same headroom and pattern width (fin width) which are presumed to be fabricated by similar lithography and dry etching processes. Results show that the stacked nanosheet structure has advantages only when the intersheet spacing or vertical sheet pitch is less than the sheet width. And for nanowire transistors almost have no advantage as compared with FinFET and nanosheet in terms of size scaling. Considering the technological constraints for the gate oxide and metal gate thicknesses, the minimum intersheet spacing should be in the range of 7 to 8 nm. Then, the nanosheet structure has advantages in boosting the chip density as compared to the FinFET counterpart only when the sheet width or fin width is wider than 8 nm. In addition, considering the process complexity, larger mobility degradation, and fringing capacitance, the nanosheet transistor does not have many advantages as compared with the FinFET ones for a footprint width less than 10 nm.

## Figures and Tables

**Figure 1 nanomaterials-12-01739-f001:**
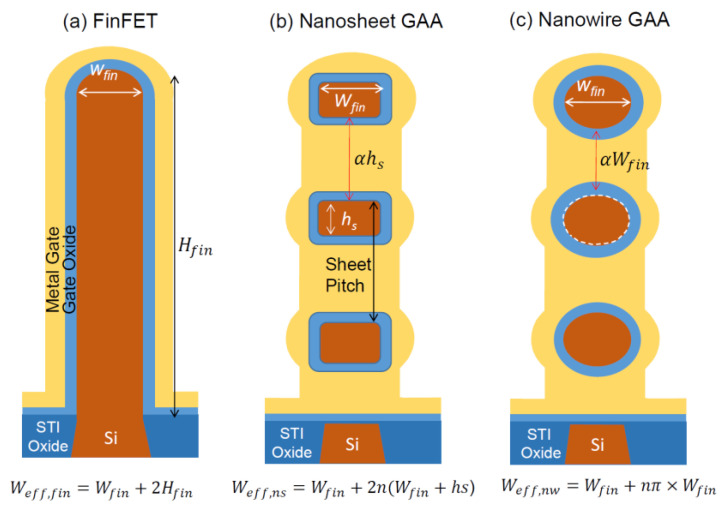
Comparison of (**a**) FinFET, (**b**) vertically stacked nanosheet GAA FET, and (**c**) vertically stacked nanowire GAA FET with the same footprint width of *W_fin_* and fin height of *H_fin_*. The formulae in the bottom show the overall effective gate width of the transistors. The numbers of nanosheets and nanowires depend on the sheet/wire sizes and the intersheet or interwire spacing.

**Figure 2 nanomaterials-12-01739-f002:**
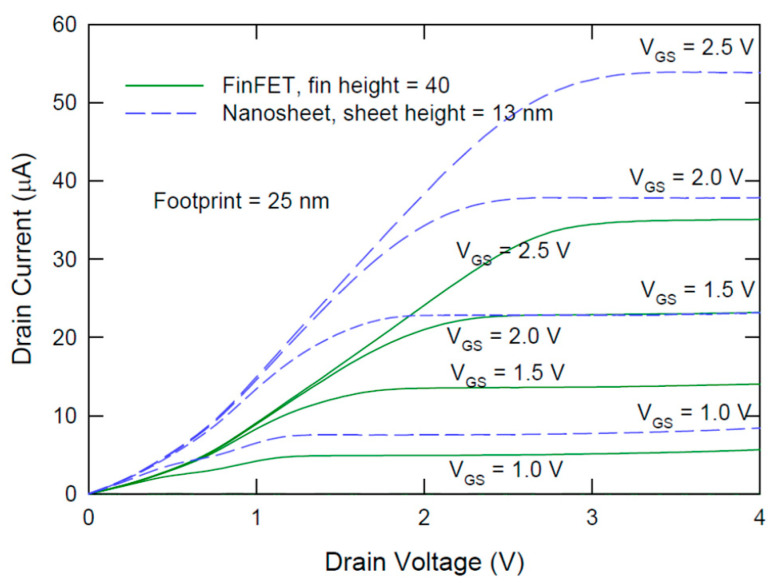
Comparison of experimental current–voltage characteristics of FinFET and nanosheet transistor with same footprint and headroom. The gate length of the transistors was 2 µm.

**Figure 3 nanomaterials-12-01739-f003:**
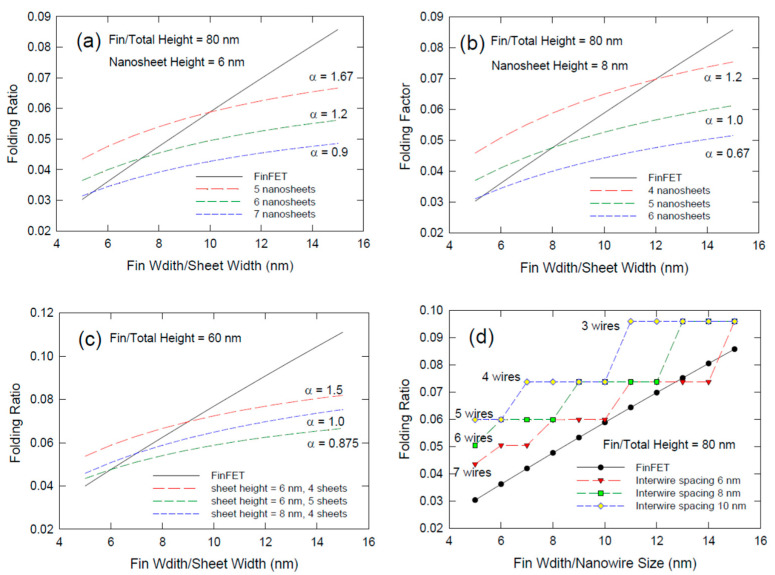
Comparison of the folding ratios as a function of fin width for different fin heigh, and different numbers of nanosheets/nanowires: (**a**) fin height = 80 nm, sheet height = 6 nm, different numbers of sheets are achieved by using different intersheet spacings; (**b**) fin heigh = 80 nm, sheet heigh = 8 nm; (**c**) fin height = 60 nm; and (**d**) different numbers of nanowires for VNWFET.

**Figure 4 nanomaterials-12-01739-f004:**
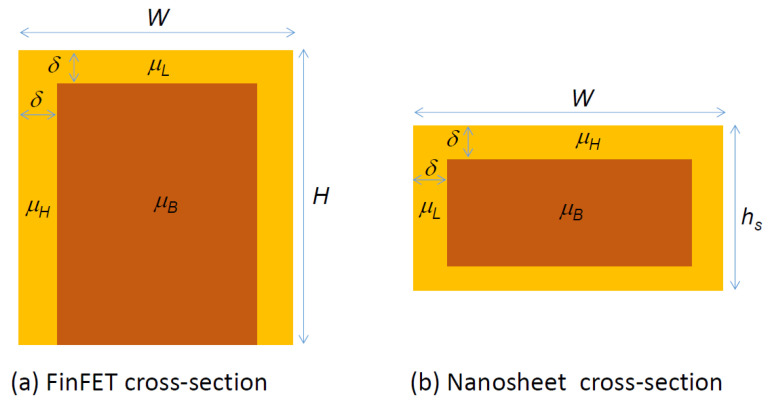
Cross-sectional view of (**a**) FinFET and (**b**) a single nanosheet with the best crystal orientation alignment for maximizing the current conduction for the high surface mobility, *µ_H_*, and low surface mobility, *µ_L_*. *δ* represents the rough surface layer which could lead to a significant mobility degradation as the size of the fin or nanosheet become smaller.

**Figure 5 nanomaterials-12-01739-f005:**
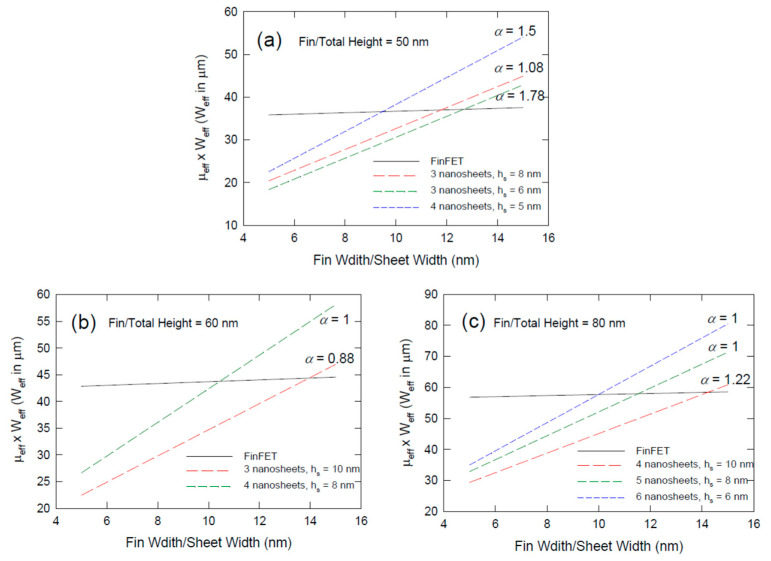
Plots of effective mobility-effective channel width product as a function of fin width or sheet width for FinFET and VNSFET for headroom of (**a**) 50 nm; (**b**) 60 nm; and (**c**) 80 nm.

**Figure 6 nanomaterials-12-01739-f006:**
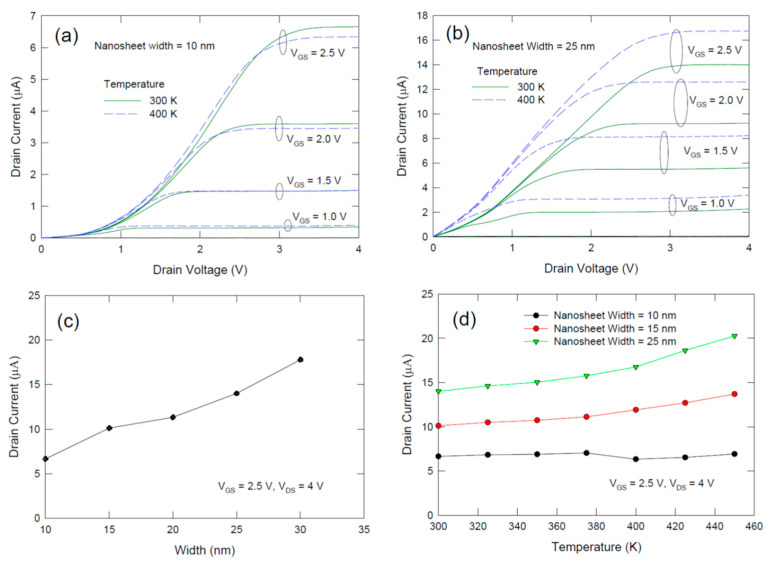
(**a**) Experimental current–voltage characteristics of a 10 nm width single layer nanosheet FET measured at 300 and 400 K; (**b**) current–voltage characteristics of 25 nm width nanosheet FET measured at 300 and 400 K; (**c**) plot of drain current as a function of sheet width; (**d**) temperature dependence of drain current for nanosheet FET with different sheet width. The gate length of the device was 2 μm.

**Figure 7 nanomaterials-12-01739-f007:**
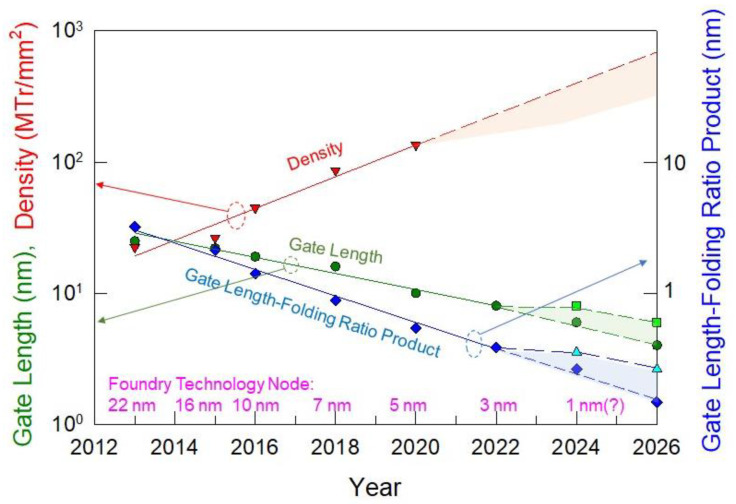
The upscaling of chip density still follows Moore’s law with the continued reduction in the layout size (redrawn based on [18]). The gate-length-reduction rate was slowed down. Additional footprint reduction was achieved by folding the channel width around the tri-gate FinFETs and by increasing the FinFET for 5 nm technology and beyond.

## Data Availability

Data sharing is not applicable to this article.

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
