# Peer review of "On the Vertically Stacked Gate-All-Around Nanosheet and Nanowire Transistor Scaling beyond the 5 nm Technology Node"

_nanomaterials, 2022, doi:10.3390/nano12101739_

Round 1
Reviewer 1 Report
In the paper, the authors perform a detailed comparison of the channel width folding effectiveness of the FinFET, vertically-stacked nanosheet transistor (VNSFET), and vertically-stacked nanowire transistor (VNWFET) under the constraints of the same vertical (fin) height and layout footprint size (fin width) defined by the same lithography and dry etching capabilities of a foundry. The authors have conducted an intuitive and comparative analysis on the scaling issue of Fin-FETs and vertically-stacked nanosheet transistors, and vertically-stacked nanowire transistors beyond the 5 nm technology node.
The reviewer has few notes which should be commented or amended by the authors to better reveal the manuscript results.
1) The left axis of the Fig.7 depicts two values, which are “Gate length” and «Density». However, it is not clear, if numeric values on the left axis are applied to both “Gate length” and “Density”. The reviewer suggests to fix Fig. 7 accordingly or describe the axis designations more detailly. Besides, the reviewer recommends to clarify the curves depicted by “squares” and “triangles”.
2) In the paper it is desirable to extend the comparison of FinFET, vertically-stacked nanosheet transistor, and vertically-stacked nanowire transistor technologies and find the most commercially advisable manufacturing technology.
3) It would be greatly interesting to compare reliability of the FinFET, vertically-stacked nanosheet transistor, and vertically-stacked nanowire transistor and ability of those to function under variety of stress modes such as high fields, radiation, etc.
Author Response
In the paper, the authors perform a detailed comparison of the channel width folding effectiveness of the FinFET, vertically-stacked nanosheet transistor (VNSFET), and vertically-stacked nanowire transistor (VNWFET) under the constraints of the same vertical (fin) height and layout footprint size (fin width) defined by the same lithography and dry etching capabilities of a foundry. The authors have conducted an intuitive and comparative analysis on the scaling issue of Fin-FETs and vertically-stacked nanosheet transistors, and vertically-stacked nanowire transistors beyond the 5 nm technology node.
The reviewer has few notes which should be commented or amended by the authors to better reveal the manuscript results.
- The left axis of Fig.7 depicts two values, which are “Gate length” and «Density». However, it is not clear, if numeric values on the left axis are applied to both “Gate length” and “Density”. The reviewer suggests to fix Fig. 7 accordingly or describe the axis designations more detailly. Besides, the reviewer recommends to clarify the curves depicted by “squares” and “triangles”.
Reply/Action Taken: Thanks. The figure is modified as advised and explanations to “squares” and “triangles” are added. See lines 515-517.
- In the paper it is desirable to extend the comparison of FinFET, vertically-stacked nanosheet transistor, and vertically-stacked nanowire transistor technologies and find the most commercially advisable manufacturing technology.
Reply/Action Taken: Yes. It is a good suggestion and the author believes that will provide a clearer strategy and direction on the further technology scaling of the foundries. Unfortunately, most of the data are still strictly confidential. We would look into this possible in the future.
3) It would be greatly interesting to compare reliability of the FinFET, vertically-stacked nanosheet transistor, and vertically-stacked nanowire transistor and ability of those to function under variety of stress modes such as high fields, radiation, etc.
Reply/Action Taken: The present work focuses on the scaling issue to sustain Moore’s Law scaling for higher density. Reliability issues such as high-field stressing are of vital importance to the gigascale circuits. It is worth tons of comprehensive investigation. In our next work, we shall try to compare the electric field distribution in VNSFET and FinFET in different sizes via TCAD simulation and modeling to further investigate their impact on scaling. Thank you very much for this nice suggestion.
Reviewer 2 Report
Interesting comprehensive study on the effective channel width of FinFET, VNSFET, and vertically-stacked nanowire transistor (VNWFET) with the same confinement of photolithography-defined layout footprint, and the etching-governed vertical height or headroom of the silicon nanostructure.
Although simple inapproach of effctive width, the work brings some insight in the W effect in various technologies.
But the paper requires major revisions:
1) How the Id-Vd curves are calculated in Fig 2 and Fig. 6 ?
2) I dont understand the non ohmic behavior of Id-Vd curves at small Vd ?
3) The channel length is not mentionned in any cases!
4) The voltages for Vg and Vd are very large for advanced technologies where for such advanced nodes the Vdd should be below 1V!
Could you explain all these issues?
Author Response
Although simple inapproach of effctive width, the work brings some insight in the W effect in various technologies.
But the paper requires major revisions:
1) How the Id-Vd curves are calculated in Fig 2 and Fig. 6 ?
Reply/Action Taken: They are measurement data. Sorry for being not made it clear in the previous version. We add “experimental” to several places in the text and figure caption.
2) I dont understand the non-ohmic behavior of Id-Vd curves at small Vd ?
Reply/Action Taken: The non-ohmic behavior at small drain voltage should be due to the more predominant effects of the surface roughness scattering and surface defect trapping which results in smaller channel mobility and a smaller amount of charge carrier in the channel. We have added this explanation to the text. See lines 450-455. Thanks.
3) The channel length is not mentioned in any cases!
Reply/Action Taken: The gate length was given in the sample fabrication section. It is 2 um. We add the description to the figure captions now. See line 429.
4) The voltages for Vg and Vd are very large for advanced technologies where for such advanced nodes the Vdd should be below 1V!
Reply/Action Taken: The gate length is 2 um long and so Vdd is large. The device was made in the university laboratory facility and does not have the capability for better gate patterning and drain/source implant control for shorter gate lengths. Anyway, we focus on the width effect only.
Round 2
Reviewer 2 Report
The authors have complied with my remarks and the revsied version is good enough